# Status Quo on Graphene Electrode Catalysts for Improved Oxygen Reduction and Evolution Reactions in Li-Air Batteries

**DOI:** 10.3390/molecules27227851

**Published:** 2022-11-14

**Authors:** Ganesh Gollavelli, Gangaraju Gedda, Raja Mohan, Yong-Chien Ling

**Affiliations:** 1Department of Humanities and Basic Sciences, Aditya Engineering College, Surampalem, Jawaharlal Nehru Technological University Kakinada, Kakinada 533437, India; 2Department of Chemistry, Presidency University, Banglore 560064, India; 3Department of Chemistry, National Tsing Hua University, Hsinchu 30013, Taiwan

**Keywords:** Li-O_2_, battery, graphene, electrodes, catalysts

## Abstract

Reduced global warming is the goal of carbon neutrality. Therefore, batteries are considered to be the best alternatives to current fossil fuels and an icon of the emerging energy industry. Voltaic cells are one of the power sources more frequently employed than photovoltaic cells in vehicles, consumer electronics, energy storage systems, and medical equipment. The most adaptable voltaic cells are lithium-ion batteries, which have the potential to meet the eagerly anticipated demands of the power sector. Working to increase their power generating and storage capability is therefore a challenging area of scientific focus. Apart from typical Li-ion batteries, Li-Air (Li-O_2_) batteries are expected to produce high theoretical power densities (3505 W h kg^−1^), which are ten times greater than that of Li-ion batteries (387 W h kg^−1^). On the other hand, there are many challenges to reaching their maximum power capacity. Due to the oxygen reduction reaction (ORR) and oxygen evolution reaction (OES), the cathode usually faces many problems. Designing robust structured catalytic electrode materials and optimizing the electrolytes to improve their ability is highly challenging. Graphene is a 2D material with a stable hexagonal carbon network with high surface area, electrical, thermal conductivity, and flexibility with excellent chemical stability that could be a robust electrode material for Li-O_2_ batteries. In this review, we covered graphene-based Li-O_2_ batteries along with their existing problems and updated advantages, with conclusions and future perspectives.

## 1. Introduction

The lucrative battery industry is expected to develop at a 14.1% compound annual growth rate (CAGR) from 2020 to 2027 as a result of the energy demands driving it. Among the batteries, lead-acid, lithium-ion and nickel metal hydride batteries are the most common. Due to the huge competition in the market among the manufacturers, the prices of batteries have also been declining along with their improved performances [1]. Seeing that the pressing needs of the alternative energy source are going high, it is indispensable to look for substitutes to fulfill the future demands.

Metal ion batteries from s-block (groups IA and IIA) and some d-block elements are well-adopted as electrode materials for rechargeable batteries. This is due to its redox potential, molecular weights, ionization potentials, ionic conductivity, and the tendency of loss and gain of the electrons reversibly under suitable currents or electrochemical conditions. The IA group metals such as Li, Na, and K are good candidates in their univalent state [2,3,4,5,6]. The IIA metals such as Mg and Ca in their divalent state and Al in a trivalent state are also good for batteries due to their high electron throughput, theoretical energy densities and better safety than Li batteries [7,8,9,10]. Other d-block elements such as Ni, Zn, Cd, and Pb are also well reported [1,11,12,13].

Among all of these, Li batteries gain a lot of popularity in the current times due to their lesser weight, suitable redox potentials, producing stable voltages in real-time applications for mobiles, laptops and electronic vehicles (Figure 1). Hence, the outstanding contributions of Li^+^ battery inventors, Prof. John B. Goodenough, M. Stanley Whittingham, and Akira Yoshino were honored with the Nobel prize in 2019 [14]. Apart from its terrific success, the recycling of Li^+^ batteries also needs to be addressed clearly along with its safety, price, and environmental concerns [15,16]. However, among Li batteries, when compared with Li^+^ batteries, Li-CO_2_, Li-S, and Li-Air batteries have better theoretical power densities (Figure 1 and Table 1) [17,18,19,20]. It is important to understand the current gap between the practical and theoretical energy densities of the above cells including multivalent metal-ion batteries. The investigation about the ion transport, storage mechanism, and electrode materials has to intensify to nexus the gap quickly [11].

Compared to the most successful Li-ion batteries [21,22], the Li-O_2_ batteries are expected to produce high densities of power and to result in a projected > 5.5% CAGR of global share from 2022 to 2027 [23]. However, the present Li-O_2_ batteries have many drawbacks, such as low stability, short life cycles, and poor energy efficacies. Hence, it is imperative to improve the existing challenges in Li-O_2_ batteries and to reach the expected power densities practically [23,24]. Apart from the Li-O_2_ batteries, several metal-air batteries such as Al-Air, Mg-Air, Fe-Air, and the Zn-Air battery systems have also been under research investigation to elevate their intriguing properties [25,26,27]. Among the battery components, stable electrode materials are the main research interest at present. Several materials have also been investigated to produce stable ORR and OER at the cathode. Among them, graphene is the most recently studied one in the carbon family.

Graphene is a class of 2D material with a sp^2^ hybridized carbon network. Morphologically it looks like a sheet or a mat at stable conditions, while at unstable conditions it rolls like a mat and produces a 1D tube-like structure known as a carbon nanotube. Both these forms gained enormous popularity as high thermal and electrically conducting materials with high mechanical properties and surface areas. Among these sheets and tube-like structures, sheet-like graphene has proven to be superior and won the Nobel Prize in 2010 due to its exceptional unseen properties till 2004 by Andrew Geim and Konstantin Novoselov [28]. It is also called the strongest and thinnest material in the universe with ultra-high surface sensitivity, optical transparency, chemical stability, and flexibility. These features are significant enough for it to be a good electrode material. Owing to its high surface area and surface chemistry, numerous inorganic and organic materials can be doped or functionalized to make composite materials that offer the desired properties [29,30].

The focus of this review is Li-O_2_ batteries and the current research on graphene electrodes as cathode materials. Hence, we would like to briefly review on Li-O_2_ batteries’ fabrication, the graphene/doped graphene/metals and their metal oxide composites, along with other carbon cathode materials. Apart from that we have also highlighted the advantages and disadvantages of graphene electrodes and the challenges in Li-O_2_ batteries with future perspectives.

**Figure 1 molecules-27-07851-f001:**
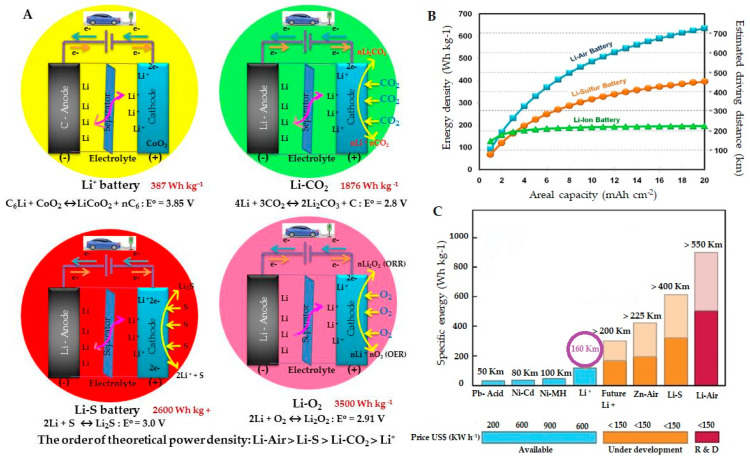
(**A**) The emerging batteries based on Li: Li^+^ battery, Li-CO_2_ battery, Li-S battery, and Li-O_2_ batteries and their theoretical power densities. Except for the Li^+^ battery, the rest of the inventions are still at the R & D level. (**B**,**C**) are different rechargeable batteries with their specific energies, areal capacity, and their mileages along with their status of prices and development. Reprinted/adapted with permission from Refs. [31,32]. Copyright 2020, copyright SCI Reports, and Copyright 2017, copyright MDPI.

## 2. Li-O_2_ Batteries

Typically, Li-O_2_ batteries belong to second-generation batteries which are known to be rechargeable (See Table 1 for different types of Li-based second generation batteries and their cell constituents). The cell contains four major components: (1) the anode, (2) the cathode, (3) the electrolyte, and (4) the separator. Here in the cell, the discharging and charging process takes place by reversible oxidation and reduction reactions at the anode and cathode. During oxidation at the anode, the Li converts into Li^+^ and gives one electron (Li → Li^+^ + e^−^). The electrons through the external circuit and Li^+^ via electrolyte reach the cathode to reduce the O_2_ into O_2_
^2−^ and to produce Li_2_O_2_ (lithium peroxide). This process is called the ORR (2Li^+^ O_2_ → Li_2_O_2_), whereas in the charging process the reverse is true (Li_2_O_2_ → 2Li^+^ + O_2_ + 2e^−^). In this case, the O_2_ is evolved at the cathode and is called OER [3]. These two ORR and OER are very crucial in the continuous power supply in the cell (2Li^+^ + O_2_ ⇌ Li_2_O_2_) as shown in Figure 2A. The first discharge and charging curves corresponding to the ORR and OER are shown in Figure 2B for Li-O_2_ cells [20]. Hence, the nature of the anode, cathode, and electrolyte materials plays a distinctive role in stable, long life, and highly efficient Li-O_2_ batteries [24].

Cell reaction:

Anode: Discharge

Li^+^_(s)_ + e^−^ + O_2(g)_ → LiO_2(s)_

LiO_2 (s)_ + e^−^ + Li^+^_(s)_→ Li_2_O_2(s)_

Half-cell reaction (ORR): 2Li^+^_(s)_ + 2e^−^ + O_2(g)_ → Li_2_O_2 (s)_

Cathode: Charge

Li_2_O_2(s)_ → LiO_2(s)_ + e^−^ + Li^+^_(sol)_

LiO_2(s)_ → O_2(g)_ + e^−^ + Li^+^

Half-cell reaction (OER): Li_2_O_2(s)_ → O_2(g)_ + 2e^−^ + 2Li^+^

Overall cell reaction (ORR/OER): Discharge/Charge

2Li^+^_(s)_ + 2e^−^ + O_2(g)_ ⇌ Li_2_O_2(s)_

Cell Potential E^0^ = 2.96V; expected theoretical power density = 3505 W h kg^−1^

### 2.1. Anode

The anode is one of the most important electrodes in a cell along with the cathode. As discussed above, the anode performs the role of oxidation where the discharge process takes place to produce half-cell potential. Anode materials are usually fabricated by carbon materials such as graphite, CNTs, graphene, porous carbon, metals, alloys, and Si-containing compounds, etc. in Li^+^ batteries [33]. To prompt the electrode, the anode materials should be robust and active in their performance to produce better battery life. In the considerations of electrode fabrication, the material bulk analysis, surface interaction with the electrolyte and other electrode components, conductivity, ion mobility, charge/discharge rates, and overall cell capacity based on it are indispensable [34]. Usually, graphite is the most versatile anode material. However, graphene could offer better performance due to its high surface area, thermal and electrical conductivity, flexibility and strength that allow it to be used in flexible electrodes. It can host more Li^+^ yet is inert while serving its role both as an anode and cathode [35]. On the contrary, in Li-O_2_ and Li-S batteries, the anode is Li metal, as it has to produce higher areal power densities than graphite in Li-ion batteries to fulfill the high energy needs, whereas the graphite anode limits the power densities to 372 mA h g^−1^, which is 10 times lesser than the Li anode in Li-ion batteries [31]. Though the Li-O_2_ battery has advantages, it still has to overcome some of the problems at the anode and cathode. The unstable solid electrolyte interface (SEI), anode pulverization by LiOH accumulation, dead Li growth and dendrite growth, and cyclic volume expansion of the anode have to be addressed clearly for stable, high-performance batteries [36]. To solve the problems at the anode, various attempts are made. Those are the usage of SEI stabilizing additives for the modification of the electrolyte for dendrite prevention and using separators, polymers, and solid electrolytes.

### 2.2. Cathode

The cathode is where reduction takes place in a cell. It is usually made up of metal oxides, metals, porous carbon, graphene, CNTs, and their composites. In the case of Li-O_2_ batteries, the cathode is O_2_, which can get absorbed from the air to the respective electrode materials and trigger the ORR during the discharge process. This reaction produces Li_2_O_2_. As these cells are rechargeable, the OER takes place during the charging process at the same electrode and it releases Li^+^, O_2_, and e^−^. Most of the experiments involved use pure oxygen, and in real-time work the air also contains different gases such as CO_2_, N_2_, and H_2,_ which are considered cathode gases. Owing to their high surface area and the porous nature of carbon electrodes, the air/O_2_ diffuses into it and gets reduced into Li_2_O_2_ by combining with the Li ions from the anode. Although this process is expected to produce high power density, some of the cathode challenges still have to be resolved. Those are the unexpected side reactions at the electrode and electrolyte. The exact formation of the Li and O_2_ products such as Li_2_O and LiO_2_ for reversible Li_2_O_2_ formation should be known. To tackle these problems of cathode overpotential due to the Li and O_2_ reactions, several research attempts are made to smooth the reversible reaction of Li_2_O_2_ formation and decomposition. In this direction, noble metals, transition metal oxides, porous nanomaterials, and carbon composites have been investigated to improve battery life and efficiency [37,38,39,40,41,42,43].

To provide the solutions to the above anode and cathode problems, the electrodes are fabricated with reduced graphene oxide (rGO) aerogel (GA) containing microspheres of hollow NiCo_2_O_4_ (NCO). The anode made up of Li infusion has revealed a significant reduction in dendrite growth and volume expansion. The power capacity near 3400 mA h g^−1^ has obtained to this electrode [36]. The Li@GA anode has shown greater cycling capacity (700) and stability than the bare Li anode. The NCO@rGA cathode also revealed the diminished overpotential to uplift the energy efficacy of the cell by providing more than 400 cycles of charging and discharging by excellent OER of NCO [37]. Figure 3 shows a flexible Li-O_2_ battery, where both the anode and cathode have been functionalized with GO/rGO with NCO. The battery has an excellent performance in different shapes: folded, curved, and reflattened. It infers that the anode (Li) and cathode (NCO and O_2_) are being protected with GO and rGO from its degradation or volume expansion for facile stable performance.

### 2.3. Electrolyte

The nature of electrolytes has an immense influence on the electrode performance and stability to produce a consistent long-range power supply in an electrochemical cell. An electrolyte is the ionic conduction medium between the two half cells such as the anode and catholic compartments to complete the cell reaction. The nature of the electrolyte and its composition has a great effect on cell performance and stability. The electrolytes could be aqueous, non-aqueous, ionic, or solid. Based on the nature of electrolytes, the Li-Air battery has been classified into four different types: (1) an aprotic Li-O_2_ battery, (2) a protic/aqueous Li-O_2_ battery_,_ (3) a hybrid (aprotic/aqueous) battery, and (4) a solid-state Li-O_2_ battery [44,45].

#### 2.3.1. Non-Aqueous/Aprotic/Organic Li-O_2_ Batteries

In this type of cell, the electrolyte contains non-polar solvents, Li salts, and other components. These cells were the first reported Li-O_2_ batteries in 1996 by Abraham and Jiang [46]. After a decade, in 2006 Ogasawara et al., reported LiPF_6_ in propylene carbonate (PC) electrolyte, and these materials have shown very good cycling ability in the Li-O_2_ cells [47]. However, the LiPF_6_ PC electrolyte prone to decomposition was observed. Hence, several electrolyte solvents are proposed, such as dimethoxyethane (DME), dimethyl sulfoxide (DMSO), and tetraethylene glycol dimethyl ether (TEGDME) with the salts of LiPF_6_, LiAsF_6_, LiSCO_3_CF_3_, LiN(SO_2_CF_3_)_2_, LiFSI, LiTFSI. However, It was claimed that room temperature ionic liquids (RTIL) 2,2,6,6-Tetramethylpiperidine-1-oxyl (TEMPO) along with solvents related to ethers, amides, and esters are more stable than the carbonate electrolytes. After that, redox mediators (RM, ex- N-methyl phenothiazine) are observed to produce stable cyclic performance followed by cellulose Kimwipes^®^ papers which can also limit the Li dendrite formation. The organic electrolytes are considered safe as these do not interact with Li electrodes as protic solvents do. Moreover, these electrolytes favor the recharging process very well [44,48,49,50,51]. See Figure 4 for the corresponding electrolyte in Li-Air batteries. In the discussion about electrolytes in the literature, the very minor changes in the electrolyte cause a great influence on the cell mechanism and affect battery performance [50,51]. The ideal characteristics of the electrolyte are to have good chemical and electrochemical stability, inertness/stability towards oxygen, and to promote the reversible decomposition of Li_2_O_2_. Hence, the best one yet to find for the fulfillment of the ideal characteristics of the electrolytes in Li-O_2_cells.

The typical cell reaction of aprotic electrolytes (Ex: RTIL) used in the Li-O_2_ cells is
2Li^+^ + RTILs + O_2_ + 2e^−^ ⇌ Li_2_O_2_ + RTILs (Deposition + decomposition)

#### 2.3.2. Aqueous Electrolyte Li-O_2_ Battery

The cell of aqueous Li-O_2_ batteries uses water, acid, and alkali components as electrolytes. The Li-Air batteries composed of these electrolyte types can generate more cell potential than organic electrolyte-containing batteries. It is due to the solubility of the discharged yields in the aqueous electrolyte medium than organic electrolytes. Because of this, there is a decrease in the membrane and electrode pore damage and diminished cathode polarization by improved O_2_ diffusion. However, in the case of organic electrolytes, the discharged products do not dissolve in the media of the electrolyte, leading to the anticipated electrode damage and low cell voltages. The same clogging phenomenon is also observed in aqueous electrolytes at higher pH and ambient temperatures. It is presumable that the forward reaction product, LiOH, did not dissolve in alkali conditions (12.8 g/100 g H_2_O) at given temperatures and used to be in the form of LiOH·H_2_O.The problems caused by precipitation can be controlled by maintaining the pH of the electrolyte by adding buffer solutions [52,53]. Another problem of alkaline electrolytes is the formation of LiCO_3_ during the cathodic reaction of Li^+^ with CO_2_ instead of O_2_. This can be prevented by adding soda-lime filters and anion membranes around the cathode [54,55,56].

The cell reaction of aqueous Li-Air battery:

½ O_2_ + 3H^+^ + 2e^−^ ⇌ H_3_O^+^ or H_2_O + H^+^ (Acidic)

½ O_2_ + H_2_O + 2e^−^ ⇌ 2OH^−^ (Alkali)

2Li^+^ + ½ O_2_ + 2H_2_O + 2e^−^ ⇌ 2LiOH + H_2_O (or) 2LiOH·H_2_O (ppt)

The following are examples of aqueous acidic and basic electrolytes:

Acidic: CH_3_COOH, H_3_PO_4_, HCl and H_2_SO_4_

Alkali salts: LiOH, LiClO_4_, LiNO_3_, and LiCl

#### 2.3.3. Solid-State Li-O_2_ Battery

There are some noticeable problems associated with protic and aprotic electrolytes that have been identified. These are due to the electrolyte evaporation at elevated cell temperature and electrolyte interaction with electrode materials leading to dendrite formation. These problems can be nullified by opting for solid electrolytes in the cell manufacturing process because of their non-volatility, no cell leakage, and high Young’s modulus. The adaptation of this specific type of electrolyte can overcome the typical electrode problems originating from the air (O_2_, CO_2_, and H_2,_ etc.) and from moisture. Typically, solid electrolytes are considered polymer and inorganic types. The very best examples of solid inorganic electrolytes are NASICON-type LATP (Li_1+x_Al_x_Ti_2−x_(PO_4_)_3_)/LAGP (Li_1+x_Al_y_Ge_2−y_(PO_4_), perovskite, antiperovskite and, garnet-type oxides. These can overcome the limitation of poor Li^+^ conductivity, hence are considered as good materials for Li-O_2_ battery fabrication. The polymer types are Li salts of PEO (polyethylene oxide), PAN (polyacrylonitrile), PVDF (poly(vinylidene fluoride)), PMMA (poly(methyl methacrylate)), PVDF-HFP, and PTFE (poly(tetrafluoroethylene)), and are extensively adopted in Li-Air batteries. However, these materials suffer from low Li^+^ conductivity due to the crystallinity of polymers, poor oxygen diffusion, and decomposition hindered charge and discharge process. Hence, inorganic solid electrolytes are considered the best for this purpose [50,57,58,59]. The expected features of the solid-state electrolyte for a high-performing Li-Air battery should follow the following criteria: (1) economic fabrication, easy device incorporation, and eco-friendliness, (2) good electrode compatibility with chemical inertness, (3) high mechanical and thermal stability during the cell reaction, and (4) ability to tolerate with Li^+^ and O_2_ species (Li_2_O_2_ and L_2_O) generated during the reaction at the cathode.

#### 2.3.4. Hybrid (Aprotic/Aqueous/Solid) Li-O_2_ Battery

The hybrid type contains aprotic and aqueous electrolytes to overcome the aforementioned problems. Here the Li anode region is fixed to contact with an aprotic electrolyte and the cathode environment is made to contact with an aqueous electrolyte. This kind of system can minimize the dendrites at the anode and avoid the lithium oxides and hydroxides deposit to block the pores of the cathode electrode. There is a LISICON (Lithium-ion Superionic Conductor) membrane to separate these two compartments to facilitate the ion transport safely. The water-stable ceramic LISCON can avoid Li corrosion damage from water and air. Hence, the drawbacks of the individual electrolyte can be conquered by this design of batteries [60]. According to the survey of Manthiram et al., the theoretical power densities of the aprotic electrolytes are the same as hybrid electrolyte (solid-liquid) Li-Air cells. Among the acid-solid and aqueous-solid cells, the theoretical energy densities are higher for aqueous-solid electrolyte cells due to the high molecular weight of the Cl^−^ in the HCl electrolyte cells [61]. It was also revealed that the aqueous- solid type can overcome the problems associated with organic and solid type electrodes [60]. The concepts of hybrid electrolytes are greatly advantageous to improvising cell performance by taking care of individual electrolyte drawbacks [62,63].

As electrolytes play a key role in battery performance, researching advanced electrolytes currently underway. Eutectic electrolytes are new materials that started attracting researchers’ attention recently. These are made by mixing a certain ratio of solid N-methylacetamide and lithium bis(trifluoromethanesulfonyl)imide. The cell fabricated by these types of electrolytes has shown a high discharge capacity (8647 mA h g^−1^) and cycling lifetime (280 cycles). Owing to the high ionic mobility/conductivity, good thermal and electrochemical stability, and compatibility with the Li anode, these eutectic electrolytes can overcome the anticipated problems associated with the existing electrolytes [64,65].

### 2.4. Separator

The separator made from polymer materials can directly separate the anode from the cathode, yet permits the electrolyte/ions from either side of the electrodes. The separator avoids the cell from absorbing the non-solid electrolyte and prevents short-circuiting. In general, the Li anode forms the dendrites which can pierce into the separator and disturb the cell. The polymers are usually replaced by ionic conductive polymers and graphene oxide (GO). These two types of electrodes can help to protect the electrode from corrosion and improve ionic conductivity [66]. The separators can act as a moderator to prevent the blast due to the ion acceleration between the anode and cathode. The most adopted separators are polyolefin (PO) polymers such as polyethylene (PE), and polypropylene (PP). The emerging separators are ceramic blended “wet” PE membrane, ceramic/polymer coated PO membrane, nanofiber separators, cellulose/polymer paper, ceramic/PVDF cast or sprayed layer, ceramic filled nonwovens, PET nonwoven separators, etc. [66,67,68].

In general, the separators have problems with porosity, shrinkage, penetration resistance, and meltdown due to the heat and wettability. These shortcomings should be overcome in the well-performing battery separators such as ceramics and IO (an inverse opal-inspired, seamless nanoscaffold structure) separators for improved workability [69]. To best describe the performance of the IO separators against traditional polymer separators such as PP/PE, Kim, et al. gave an excellent demonstration, as shown in Figure 5. Here the researchers fabricated the IO separator based on optimized SiO_2_ NPs, a polymer matrix of UV-cross-linked ETPTA (ethoxylated trimethylolpropane triacrylate), incorporated with a PET (polyethylene terephthalate) nonwoven substrate which helps as an amenable thermomechanical framework. This kind of nano scaffold of battery separators prepared by the inverse opal-inspired process has a more well-ordered structure than other separators (Figure 5c, right side blue color hexagonal arrays). The well-ordered structure can significantly improve the Li^+^ transfer. Figure 5A,B show the comparative electrochemical performances of IO separators with conventional PP/PE/PP separators. The IOs have given the best cycling charge/discharge potential (Figure 5A) with high stability (Figure 5B). Figure 5C is the time-of-flight secondary ion mass spectrometry (TOF-SIMS) surface analysis, and ion mapping reveals that the IO separators have less Li_2_F^+^ than the PP/PE/PP (Figure 5D). The LiF salt is from the decomposition of LiPF_6_ electrolyte in the cell. This data confirms that the IOs are highly stable, less prone to chemical reactions, and are more suitable to pass the Li^+^ toward the cathode. On the other hand the polymer separator’s surface has a greater amount of L_2_F^+^ due to its less stable and highly disordered pores in the membrane to withstand the cell reaction and ion transport [69].

In addition to IO separators, other smart battery separators have been prepared based on functionalized nanocellulose-integrated heterolayered nanomats. In a beautiful comparison with PP/PE separators, the cellulose smart cell separator displayed a better battery performance [70].

### 2.5. Electrode Membrane

It is another significant component in the batteries that helps to protect the electrode and promote the charging process by quick decomposition of Li_2_O_2_ into respective Li^+^ and O_2_ for OER reversibly. Figure 6 is the best example to explain the important role of the membranes to guard the electrodes. In Figure 6A, the Ru catalyst is supported on top of the carbon material. During the discharge process, the Li^+^ reaches the cathode surface from the anode through the electrolyte and is reduced into Li_2_O_2_ and some other Li oxides such as Li_2_O. These oxides form an insulating layer around the cathode and passivate Ru. Hence, there is a lazy OER during the charging process, and it will lower the cycling capacity and battery life. To avoid this scenario, a catalytic membrane that can act as a mask is necessary to avoid direct electrode contact and to also avoid electrode passivation or corrosion. Figure 6B illustrates an attempt of Pt NPs loaded PAN NF (polyacrylonitrilenanofiber) membrane which can help to prompt the OER by reserving the Ru active sites on the support and maintenance of the co-catalyst mechanism. Such how the membrane could provide additional catalytic sites and decompose the insulating Li_2_O_2_ and Li_2_O into Li^+^ and O_2_. As a result, the charging process will be accelerated, and the cell cycling efficacy (60 cycles) and capacity is improved by about 1000 mA h g^−1^ compared to the non-protected cathode [71,72]. Apart from this work, the cathode made up of encapsulated RuO_2_ NPs in N-doped graphene also has shown improved cyclic performance and battery stability to bare RuO_2_ catalyst [41]. Here the doped graphene is presumed as a cathode-protecting membrane.

## 3. Graphene and its Composite Catalysts as Electrodes

The designing and fabricating of efficient, stable, safe, and economic electrode materials are indispensable in the battery industry. Graphene-based battery electrodes are among the foresaid materials with exceptional features. In this section, we are going to discuss graphene-related materials as an electrode. The graphene materials are categorized into (1) graphene/GO/rGO/non-metal doped graphene; (2) graphene functionalized with metal nanoparticles as catalysts; (3) graphene with metal oxides; and (4) other carbon nanomaterials such as porous carbon, CNTs, carbon nanofibers, carbon dots, and their composites. Figure 1 represents the graphical view of this section.

This section also provide the systematic understanding of the ORR and OER at cathode. Further, we are tried to project how the efficiency of the cell is improving from bare graphene to other nanocarbon materials upon doping/functionalizing with non-metals, metals, metal oxides along with morphology. Several methods have been made to improve the Li-Air battery’s efficacy to reach the theoretical power densities along with manufacturing costs. Hence, expensive noble metals and their oxides can be replaced with economic carbon materials such as graphene, CNTs, and CDs. However, the carbon materials alone may not provide better catalytic performance. As a result, attempting to load with the different catalysts to create extra reactive sites on top of the carbon materials is well appreciated, along with taking the consideration of its morphology and dimensions. Earlier in the discussion about graphene-related electrode materials, we briefly discussed their important synthesis and characterization methods. In the end, we have also discussed the advantages and challenges in the battery industry.

### 3.1. Synthesis and Characterization of Graphene Materials

Graphene can be prepared by several methods. These include mechanical peeling, chemical vapor deposition (CVD), chemical exfoliation of graphite with organic solvents, oxidation to give graphene oxide (GO) followed by reduction into reduced graphene oxide (rGO), solid phase intercalation, the ultra-sonication method, etc. Among them, CVD and exfoliation are the most important techniques. The CVD can produce highly pure graphene with minimal or no defects and the chemical exfoliation methods can be for high throughput synthesis. As we discussed in the introduction, graphene is a layered and atomic thin material. It can be characterized with optical microscopy (OM), atomic force microscopy (AFM), scanning tunneling microscopy (STEM), transmission electron microscopy (TEM), and Raman spectroscopy to identify the number of layers, thickness, morphology (sheets or wrinkled), and defects. See Figure 7A,B for the synthesis and Figure 7C,D for the characterization of the CVD-grown graphene and solution phase exfoliated GO and rGO [75,76,77]. Figure 7A is the representation of the discovery of graphene in 2004 to the CVD-grown graphene and other layered materials up to now, whereas Figure 7B depicts the liquid phase synthesis of GO from graphite exfoliation and reduction into rGO in aqueous and organic solvents from 1958 to now, and by various other methods reported [41]. Schiavi et al. proposed the preparation of GO- and rGO-based metal cathodes from the recycling of used batteries. The projected method is green and sustainable electrode preparation for future Li -O_2_ batteries, thus limiting the consumption of primary sources [78].

Due to the dimensionality of the graphene materials, defect-free, single-layer preparation and its characterization require very good skills. Most importantly, it can be characterized by AFM, STM, and Raman to know its thickness, morphology, nature of defects, and the number of layers (Figure 7C,D). We know that single-layer defect-free graphene is highly conductive, transparent, sensitive, and possesses a high surface area. X-ray photoelectron spectroscopy is used to identify the nature of carbon and oxygen functional groups in graphene, GO, and rGO. In terms of chemistry, the graphene contains C=C and alkene type bonds exclusively, and lesser amounts of C-C alkane carbons. The oxidized form of graphene (GO and rGO) possesses many oxygen functional groups such as alcoholic/phenolic (-OH), aromatic ethers (C-O-C), aldehydes/ketones (C=O), and acid/ester/lactones (-COOH (R)). These functional groups play a crucial role in the graphene chemical properties for many reactions by functionalizing a variety of molecules on top of it covalently and non-covalently [79,80,81,82,83]. As a result, it was a versatile material in nanotechnology to adopt in many applications and especially electrode materials in batteries.

### 3.2. Graphene and Non-Metal Doped Graphene

#### 3.2.1. Graphene

Graphene (Figure 8A) [84] and its important relatives GO and rGO possess a unique surface area, electrical and thermal conductivity, and chemical robustness that have been tested as electrode materials to facilitate the ORR and OER. In the beginning, Wang et al. reported the multi-layered graphene from a pencil drawing on the ceramic separator and measured its capacity and voltages. The results are communicated that the graphene could be a cathode material in the aprotic Li-O_2_ battery [85]. By taking advantage of the surface defects and edges as reactive sites, bare graphene (rGO) is demonstrated as a cathode material. It showed improved discharge voltages and capacities in hybrid Li-O_2_ batteries, which is comparable to the 20% Pt/Carbon black [86]. Sun et al. reported another non-aqueous Li-O_2_ cell by using RGO as a cathode material which has also given improved discharge voltages and discharge capacity [87]. Storm et al., fabricated different types of rGO by hydrazinium hydroxide rGO (HyrGO) and thermally reduced rGO (TrGO) from GO. They observed that the TrGO produced higher power densities than HyrGO. This was due to the oxidation time of GO, and the type of reduction, morphology, surface area, and functional group on each rGO [88]. Very recently, Al-Ogaili et al. synthesized rGO by the green reduction of GO with plant extract of *Salvia Officinalis.* It has provided better capacity for Li-Air batteries than compared with hydrazine and NaBH_4_ reduced GO (rGO-HZ and rGO-NB). This implies that the plant extract rGO (rGO-SE) has better ORR and OER, and could act as a sustainable electrode graphene material [89]. See Figure 8F for the galvanostatic test of the above three materials with 10 mV of differences. Apart from serving as a cathode, the rGO and the CVD-grown single layer graphene could work better as an anode protecting mask called SEI. Due to its remarkable hydrophobicity, chemical robustness, and high conductivity, it can avoid the formation of the Li dendrites at the anode and passivate the Li anode towards moisture and O_2,_ followed by other side reactions [90].

#### 3.2.2. Porous Graphene

Porous carbon materials have the great advantage of a hierarchal arrangement of pores which can facilitate high surface area, conductivity, and ease of oxygen/air diffusion through it. Hence, it was expected that this would improve the discharge capacity. Due to the systematic ordered nature of the porous materials, they can promote the reaction kinetics at the cathode surface in a smooth, well-ordered manner and can reduce the overpotential during ORE and OER by increasing O_2_ mass transfer. Taking the advantage of the honeycomb sheet-like structure and π electron density, porous/crumpled graphene can be obtained by the template and non-template-assisted synthesis methods. It was revealed that the ordered porous graphene materials revealed stable cyclic power densities compared to the nonporous and commercial carbon materials [91]. Kim et al., synthesized porous rGO graphene/GO composite for a stable paper-like cathode materials for Li-Air batteries and made a comparative study with plane rGO and commercial carbon. It was found that porous graphene has a very good electrode performance than a sheet-like structure [92]. Zhong et al. prepared a porous 3D graphene membrane for a stable highly efficient cathode that can facilitate better ORR and OER with very good moisture resistance and high O_2_ flow for a stable charge and discharge process [93]. Hierarchically porous graphene and holey graphene were prepared by Xiao and Lin with exceptionally higher capacities than previous reports (Figure 8B,G) [94,95]. The porous architectures are beautifully obtained along with high capacities. See Figure 8G for the electrode performance of a cathode in a Li-Air battery. Lim et al. reported an eco-friendly synthesis of porous 3D graphene spheres as an efficient oxygen catalyst from vacuum residue. The bubble-like multi-layered rGO spheres obtained by the thermal heating process provide a stable specific capacity. The porous nature of the electrode can store the Li_2_O_2_ and avoid the formation of Li_2_CO_3_. The prepared 3D graphene spheres are green and economic for the stable cathode in Li-Air batteries with a very high discharge capacity of 18,578 mA h g^−1^ than other reports [96]. Apart from the above discussions, a few more porous materials have also reportedly been discussed, and the efficacy is improved compared with the sheet-like carbon and commercially available carbon materials. Hence, we can conclude that the morphology of the graphene plays a key role in improving the efficacy of the power densities from the sheet to the porous structure [97].

#### 3.2.3. Non-Metal Doped Graphene

Designing the materials towards specific applications and recycling of spent materials is one of the most interesting and challenging strategy than inventing on a completely new material. This is because the new inventions may demand additional labor, skill, and investment. It was noted that the defected and heteroatom-doped materials have an improved performance than the plane and un-doped ones. Hence, the new graphene has been engineered by introducing defects and non-metals as dopants by the careful replacement of carbon atoms in the skeleton. The enhanced reactivity can be expected from the dangling bonds at defects and new energy level formation by doping [98,99]. In the review on catalytic materials for ORR and OER, the researchers found that both the defective and heteroatom-doped carbon materials have comparatively the same efficacy in cell performance. However, in some instances, heteroatoms such as B, N, S, and O may change the electronic environment and its spins to provide new reactive sites and energy levels of the catalytic systems, see the heteroatom doping in Figure 8C–E. This may result in lower activation energies and ease O=O bond breakage for facile ORR and stable performance of the battery system [100]. Gong et al., for the first time, published the heteroatom doping in CNTs, and, later on, many articles in this direction were listed in the research database. This work helped to replace commercial catalysts such as Pt/C [101].

According to the discussion, the cathode material could be able to accelerate both ORR and OER. During the ORR, the oxygen should more able to gain the electrons and need easy activation to convert into O_2_^2−.^ This will be prompted in B, N, or S-doped graphene, as it has the excess of electrons in graphene and acts as an *n*-type catalyst. The OER is also equally important to complete the cell reaction. Hence, to trigger the OER in a facile manner, an electron-deficient environment is indispensable, as the Li_2_O_2_ has to convert into Li, O_2_, and electrons. This can occur by the introduction of B into the graphene to serve as a *p*-type catalyst and accept the electrons from O_2_ into B vacant orbitals. Hence, the possibility of a fast and facile OER is anticipated [102,103,104,105]. Wu et al. synthesized a 3D porous B-doped rGO by using boric acid as a source of B and a pore templating agent by adopting a simple freeze-drying method. The comparative studies of rGO and B-rGO as cathode material in Li-O_2_ batteries in Figure 8H reveal that the bare rGO has 10,000 mA hg^−1^ and B-rGO has 18,000 mA h g^−1^ at 100 mA g^−1^. The highest performance of the 3D-B-rGO is due to the modified electronic environment and morphology of the graphene [106]. Shui et al., prepared porous N-doped holey graphene (N-HGr) by the thermal annealing method, and the NH_3_ is used as a source of N. The N-HGr is used for a metal, binder-free air cathode by filtration method. The porous nature and rough surface area with additional N reactive sites and excessive electrons in the N-HGr prompted the ORR and OER reactions. Hence, they reported a high discharge capacity of 17,000 mAh g^−1^, with very good stability at 800 mA g^−1^ (Figure 8I). The researchers have demonstrated that the N-HGr is stable for more than 100 cycles. The results are comparable to the commercial Pt noble metal catalysts, and far better than the rGO and reduced holey GO at the relative comparison in the given experimental conditions tested for Li-O_2_ batteries [107]. CxNy particles at N-doped 3D porous graphene have also been prepared and demonstrated for successful air cathode with 8892 mA h g^−1^ at 1000 mA g^−1^ [108].

In the assessment of both B-doped and N-doped graphene, the B-doped graphene has shown the highest capacity. Thus, it is worthy of noting that the electron-deficient graphene platform favors the OER, which accelerates the reaction kinetics in the Li-breathing cells. This implies that the reversible soft dissociation of Li_2_O_2_ is very indispensable for stable cells to enhance cyclic efficacy and reduce the overpotential. It can also explain that the ease of OER can avoid the Li_2_O_2_ decompositions at the cathode and avoid or minimize the passivation levels of the electrode. From the above conclusions, co-doped graphene with both B and N might generate enhanced capacities by the facile promotion of ORR by N and OER by B. To introduce the hetero atoms into the C backbone, it is always a better idea to have a theoretical approximation in advance [109,110]. The DFT study of Benti et al. has shown that the B and N co-doped graphene has improved performance for ORR and OER compared to pristine graphene [111]. Apart from this, several efforts have been made to find an ideal catalyst from metals and alloys containing B/N-doped graphene [112,113,114,115,116]. Phosphorous (P) doped graphene called phosphorene was hypothetically designed to study the ORR and OER theoretically, but its practical synthesis was not identified during our search [117], whereas, P-doped pinecone-derived porous hive-like carbon has been fabricated and demonstrated for the Li-O_2_ battery. It showed a remarkably high discharge specific capacity of 24,500 mA h g^−1^ over that previously reported [118]. Unlike the B and N doping and co-doping, S-doped graphene has shown a declined performance [119], whereas the N- and S-doped graphene has improved capacity over the S-doped graphene alone [120]. The S, N co-doped graphene has also been used for Li-ion batteries, metal-air batteries, and fuel cells to improve the performance of the cell [121,122,123,124]. See Table 2 for the comparative cell performance values including the references. When studying the mechanism of graphene electrocatalysis, the bare graphene has a pool of electrons, whereas the GO and rGO electron density was less. It depends on the number of π electrons, oxygen functional groups, and defects presented on it after oxidation followed by reduction. To improve the ORR and OER, it is always good to tailor the electron density. The bare graphene encourages the ORR due to the freely available π electrons from its sp^2^ carbon network. The electron density expectedly increases by adding N and S as well. However, to complete the cell reaction, the reversible OER is also very indispensable in Li-O_2_. The OER takes place by losing the electrons of O_2_^2−^ into O_2_, and as a result, electron-deficient scaffolds which can create the electron demand and act as electron traps are prominent. This seniority can be achieved by doping the III group elements such as B and Al (Lewis acids). To achieve the facile ORR and OER the V/VI and III group elements can carefully dope with the desired concentrations and reaction parameters without disturbing the graphene backbone. Ideally, careful disturbance of the uniformity of graphene by oxidation and group III elements doping creates the electron deficiency for OER. The doping with V or VI group elements can create an electron-rich environment to facilitate the ORR. According to the above discussion, the rGO has an ideal oxygen environment for better results than graphene and GO for electron shuttle between Li and O_2,_ hence many rGO-based electrodes have been fabricated than graphene or GO.

**Figure 8 molecules-27-07851-f008:**
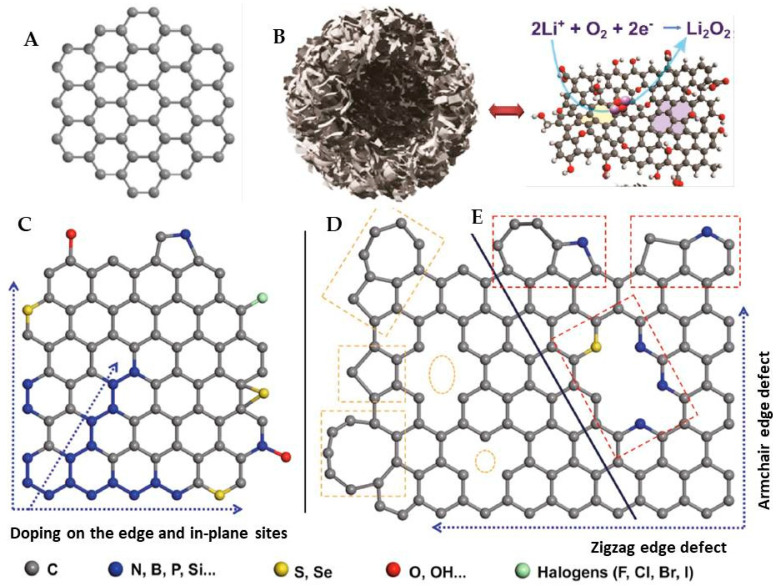
Representations of different graphene materials. (**A**) 2D plane undoped graphene [84], (**B**) 3D porous graphene with ORR in the right corner [94], (**C**) Edge and in-plane doped graphene without disturbing hexagons [100], (**D**) Defected non-doping graphene with multiple carbon rings from 5–14 carbons [100], (**E**) Defected and doped graphene with non-carbon elements such as B, N, S, P, Si, F, Cl, I, etc. [100]. The best performance of the Li-Air battery by the above materials from the galvanostatic test, (**F**) 2D graphene with different types of rGO [89], (**G**) 3D porous graphene [94], (**H**) B-doped graphene [106] and (**I**) N-doped graphene [107]. Reprinted/adapted with permission from refs. [84,94,100,106,107]. Copyright 2015, ACS. Copyright 2011, ACS. Copyright 2021, Springer. Copyright 2022, Elsevier. Copyright 2016, ACS. Copyright 2016, ACS. The experimental parameters are listed in Table 2.

### 3.3. Graphene-Metals

In this section, we discuss graphene-supported transition/noble metals as a cathode in Li-Air cells. Specifically, we conferred Ru, Fe, Ni, Co, and Mn nanoparticles (NPs) as well as their alloys in bi- and tri-metallic forms. Ru-based nanomaterials (NMs) supported on rGO used as cathodes in Li-Air cells by Jung et al. They used well-dispersed metallic Ru NPs and RuO_2_. 0.64H_2_O with 2.5 nm size and deposited onto rGO. Ru-based NPs supported on rGO were combined to create hybrid materials that effectively served as electrocatalysts for Li_2_O_2_ oxidation processes. These materials maintained cycling stability for 30 cycles without electrolyte breakdown. In particular, the RuO_2_·0.64H_2_O-rGO composite outperformed Ru-rGO in catalyzing the OER reaction, greatly suppressing the average charge potential to 3.7 V at 500 mA g^−1^ and 5000 mA h g^−1^ specific capacity [74]. The use of porous graphene material as a catalyst (see Table 2) showed noticeably larger discharge capacities compared to the layered/non-porous graphene. Additionally, among porous graphene with pores of 60, 250, and 400 nm, the medium-sized graphene pores provided the maximum discharge capacity. Additionally, Sun et al., found that the adding of nano-Ru crystals to the porous graphene enhances the OER with a high capacity, low over potential, and cyclic stability of 17,700 mA h g^−1^, 0.355 V, and 200 cycles, respectively. Hence, they concluded that the porous graphene decorated with nanoRu displayed very good cathode activity in Li-O_2_ batteries [125]. Liao et al. fabricated directly growing 5 nm porous Ru via a galvanic replacement reaction by using Ni foam as a current collector. The resulting Ru@Ni produced 3720 mA h g^−1^ capacity at 200 mA g^−1^ current density. The cathode is considered as a carbon- and binder-free material for Li-O_2_ batteries. However, its capacity is still lower than the porous graphene-supported catalyst. Hence, the reported catalyst has to be evaluated properly to enhance its performance [126]. Tan et al., reported a stable porous electrode architecture and a biphasic N-doped Co@graphene (N-Co@graphene) to a promising cathode for Li-O_2_ cells. The multiple-nanocapsule configuration enables high/uniform electroactive zones. The presence of N and Co enhances electric conductivity and catalytic activity. The prepared electrode encourages oxygen diffusion and catalytic reaction. As a result, the electrode displays significantly increased electrocatalytic qualities. [127].

Rechargeable metal-O_2_ (Air) batteries require the discovery of effective bifunctional catalysts for ORR and OER. Ren et al. used a simple releasing approach to show a direct fabrication of effective bifunctional OER/ORR catalysts made of MnNiFe/LIG (Laser Induced Graphene). It has been denoted as (LIG/M111 and LIG/M311), where the molar ratios of Mn, Ni, and Fe are denoted with numbers. Without the need for a redox mediator, the LIG/MnNiFe exhibits good performance in Li-O_2_ and Li-Air batteries. The LIG/M311 catalyst remains stable for 350 cycles, while those with the LIG/M111 catalyst only stay constant for about 300 cycles. It was also studied how the LIG/M111 and LIG/M311 catalysts in Li-O_2_ batteries work and how they affect the discharge and breakdown of products. This study highlights the effectiveness of LIG for electrode fabrication and encourages additional research into carbon-metal and oxide composite cathode catalysts for metal-air cells [128,129]. In addition to the metal-supported graphene, metal- and metal oxide-supported nanocomposite cathodes, rGO/Ru/a-MnO_2_ was prepared by the reduction and vacuum filtration method without the aid of any binder and conductive carbon for the electrode fabrication. The hybrid cathodes as well as bare rGO and rGO/Ru were produced for comparative purposes. The successfully created hybrid catalyst was shown to significantly speed up the ORR and OER. The improved performance of the electrode was due to the facile O_2_ flow between the layers of the nanocomposite designed, increasing reaction kinetics, as well as the catalytic impact of Ru and a-MnO_2_. Galvanostatic charge-discharge, CV, EIS, and electrochemical cycling tests were used to assess the cathodes’ electrochemical performance. The rGO/Ru/aMnO_2_ electrodes were demonstrated to function at full discharge capacity of 2225 mA h g^−1^, whereas rGO/Ru can supply only 1670 mA h g^−1^, owing to the synergetic effect of Ru and a-MnO_2_ catalysts [130]. However, when compared to the graphene Ru and porous graphene Ru cathodes, the hybrid rGO/Ru/a-MnO_2_ capacity is comparatively low, as tabulated in Table 3.

### 3.4. Graphene-Metal Oxides

One of the well-developed technologies for effective electrical energy storage that has been in use in the industry for many years is the battery. A highly effective ORR-based electrocatalyst is a crucial prerequisite for Li-O_2_ batteries. Noble metals, particularly Pt, display significant performance as the cathode material among all electrocatalysts for ORR [131,132]. However, the high price of Pt and its rarity prevent the use of Pt-based catalysts in commercial applications on a large scale [133,134]. As a result, major research efforts should be put into creating new non-Pt-based ORR catalysts that are highly efficient and economical. Numerous academic publications have described developing extremely active noble metal-free catalysts for ORR to date, such as carbon-based nanoparticles including graphene, carbon nanotubes, carbon nanofiber, graphene and activated carbon [135,136,137], oxides of non-precious metals, and oxides of transition metals such as Ni(OH)_2_, Fe_2_O_3_, RuO_2_, and MnO_2_ [138,139,140,141]; polyanilines and polythiophenes [142,143], and their composites [144,145,146] have been investigated thoroughly as electrocatalysts [147,148,149,150].

The skeleton of 3D graphene nanoribbons (GNRs) was obtained by synthesizing RuO_2_ with GNRs using chemical shear, and the RuO_2_ particles were then loaded using a simple dropping technique. In addition to inheriting the huge aspect ratio and 3D intersected structure from raw MWCNTs, GNRs also have a very excellent specific surface area, which increases specific capacity. According to the electrochemical results, adding RuO_2_ particles greatly lowers the charge over potential. Moreover, RuO_2_@GNRs cathodes exhibit remarkable cycling stability, with 424 cycles at 1000 mA h g^−1^. The 3D GNRs and the highly effective RuO_2_ work together synergistically to boost RuO_2_@GNRs’ catalytic performance [151]. A simple laser-induced graphene (LIG) technique is used to manufacture Co_3_O_4_/LIG, which is then used to synthesize bifunctional ORR and OER catalysts that are highly efficient. The Co_3_O_4_/LIG showed remarkable efficiency in Li-O_2_ batteries. The first 100 cycles of the cell’s galvanostatic charge/discharge profile through a cut-off capacity is 430 mA h g^−1^. At the beginning of the first cycle, the discharge/charge voltage gap was just 0.42 V. In comparison to Li^+^/Li, the discharge voltage of the Co_3_O_4_/LIG cell was 2.73 V in the first cycle and 2.67 V in the 100^th^ cycle, indicating that the cell’s energy output was stable [152]. A high-efficiency cathode catalyst is crucial for enhancing the electrochemical characteristics of Li-O_2_ batteries, particularly the cycle performance. The simple hydrothermal approach for making CuCr_2_O_4_@rGO (CCO@rGO) nanocomposites, followed by a variety of calcination procedures, is a reliable cathode catalyst. The produced CCO@rGO nanocomposites were used as the Li-O_2_ battery cathode catalyst and demonstrated exceptional cycling performance for more than 100 cycles at a static capacity of 1000 mA h g^−1^ at a current density of 200 mA g^−1^. The increased attributes were attributed to a synergistic interaction between the large specific surface area, high conductivity, and high catalytic efficiency of the spinel-structured CCO nanoparticles [153]. Furthermore, the MnO_2_ nanomaterials developed uniformly on the rGO surface to produce MnO_2_@rGO nanocomposite and were used as cathode catalysts in Li-O_2_ batteries. MnO_2_@rGO nanocomposite displayed a greater capacity of 4262 mA h g^−1^ at 100 mA g^−1^. The significant synergistic interaction between the rGO and the MnO_2_ nanomaterials on their surface is responsible for the exceptional performance. The rGO has a porous multilayer structure that fosters oxygen and ion transport, offers good electrical conductivity, and offers storage space for the discharge materials. The highly exposed surface of the nanoscale MnO_2_ facilitates the surface movement of the LiO_2_ species and prevents the buildup of discharge materials on the surface of the electrode. Additionally, throughout the discharge and charge processes, a change between non-lithiated and lithiated MnO_2_ was observed. This transition aid might be responsible for promoting electron transport between the catalyst and discharge products, hence lowering the overpotential of the oxygen evolution process [154]. A cohesive 2D/3D heterostructure NiCo_2_O_4_ (NCO)@GNS electrocatalyst was developed via ultrasonication. A porous NiCo_2_O_4_ (NCO)@GNS nanocomposite with various interfaces for catalyzing the OER as well as ORR progressions in both non-aqueous and aqueous electrolyte conditions was subsequently prepared. The dodecahedron nanosheets of NiCo_2_O_4_ with micro/meso porous structure combined with the vastly conductive graphene nanosheets to produce the NCO@GNS nanocomposite electrocatalyst. The developed nanocomposite electrode had a high discharge capacity of 7201 mA h g^−1^ at 100 mA g^−1^ in a Li-O_2_ battery. It was also demonstrated that the NCO@GNS nanocomposite had long-term cycling stability of nearly 200 cycles with charge potentials and stable discharge that were higher than those of the other evaluated electrodes. It demonstrated that the ORR and OER performance of nanocomposites were shown to be higher than that of NCO, commercial Pt/C and GNS, and catalysts. The NCO@GNS heterostructure’s high surface area, many more sites, and voids made it easier to absorb the electrolyte, adsorb oxygen, and diffuse Li^+^ ions [155]. In another work, a 3D ZrO_2_@NiCo_2_O_4_/GNS nanocomposite cathode catalyst in Li-O_2_ batteries was constructed. The Zr^4+^ ions were coated onto the nickel cobaltite matrix, exhibiting better bifunctionality for the ORR and OER. It enabled the development of three-dimensional networks for electrolyte impregnation and oxygen diffusion with ultra-less overpotential owing to Zr insertion. The 3D ZrO_2_@NiCo_2_O_4_/GNS nanocomposite’s electrochemical performance results in a greater discharge capacity of 9034 mA h g^−1^ at 50 mA g^−1^ and has a maximum capacity of 1000 mA h g^−1^ at 100 mA g^−1^, and an enhanced cycling efficiency of about 100 cycles [156]. Among all the metal oxides reported, ZrO_2_@NiCo_2_O_4_/GNS has demonstrated better electrochemical efficiency for both OER and ORR. However, these catalysts are not superior to the porous or non-metal-doped graphene as well as the metal-doped porous graphene cathodes.

**Table 3 molecules-27-07851-t003:** Comparison of the cell performances of various metal and metal oxide supported graphene electrodes.

Graphene Supported Metal Catalysts for ORR and OER						
S. No.	Catalyst	Electrolyte	Discharge Voltage (V)	Discharge Capacity (mA h g^−1^)	Current Density	Ref. No.
1	Ru-rGO	LiCF_3_SO_3_ -TEGDME	3.7	5000	500 mA g^−1^	[74]
2	Porous Graphene—Ru	LiClO_4_—DMSO	2.79	17,700	200 mA g^−1^	[126]
3	N doped Co @ Graphene	LiCF_3_SO_3_ in TEGDME	2.8	3.65 mA h cm^−2^	0.1 mA cm^−2^	[128]
4	MnNiFe/Laser Induced Graphene	LiCF_3_SO_3_/G4	2.9	26.3 mA h cm^−2^	0.08 mA cm^−2^	[129]
**Graphene supported metal oxide catalysts for ORR and OER**						
5	RuO_2_ decorated graphene nanoribbons	LiCF_3_SO_3_ in TEGDME	1.652	5397 mA h g^−1^	100 mA g^−1^	[152]
6	CuCr_2_O_4_@rGO nanocomposites	LiCF_3_SO_3_ in TEGDME	0.99	1000 mA h g^−1^	100 mA g^−1^	[154]
7	MnO_2_@rGO	LiCF_3_SO_3_ in TEGDME	0.15	5139 mA h g^−1^	100 mA g^−1^	[155]
8	3DNiCo_2_O_4_ dodecahedron nanosheets decorated@ 2D graphene nanosheets.	LiCF_3_SO_3_ in TEGDME	2.79	7201 mA h g^−1^	100 mA g^−1^	[156]
9	ZrO_2_@NiCo_2_O_4_/GNS	LiCF_3_SO_3_ in TEGDME	2.95	9034 mA h g^−1^	50 mA g^−1^	[157]

### 3.5. Other Carbon Nanomaterials (CNTs, CNFs, CDs)

Besides graphene, other forms of carbon materials also showed excellent electrochemical performance in Li-O_2_ batteries owing to their exceptional physical and chemical features [57,157,158,159]. In this part, we summarize the recent advances and trends of other nanostructured carbon materials, including one-dimensional carbon nanofibers (CNFs) and carbon nanotubes, (CNTs)/3D hierarchical architectures and their doped structures, and zero-dimensional carbon dots for Li-Air batteries.

CNTs have been extensively used in numerous energy storage devices including Li-Air batteries because of their optical, mechanical, and large surface area, excellent void volume, notable electrical conductivity, and high void volume [160,161,162]. Similar to elemental doping in graphene or fabricating graphene-based nanocomposites discussed above, doping/nanocomposites also improve the electrochemical properties of CNTs. Li et al. incorporated nitrogen into carbon nanotubes (N-CNTs) and compared it with CNT as cathode candidature in Li-O_2_ batteries [163]. Intriguingly, the N-CNTs electrodes displayed remarkably improved performance compared to the CNTs. The N-CNTs electrode has a considerable discharge capacity of 866 mA h g^−1^, whereas CNTs delivered 590 mA h g^−1^. These results indicated that N-CNTs displayed a capacity approximately 1.5 times greater than CNTs. Kwak et al. modified the carbon nanotube’s surface with well-dispersed molybdenum carbide (Mo_2_C) nanoparticles (NPs) and used them in Li-O_2_ batteries as a cathode material [164]. They observed that the Mo_2_C NPs catalyst facilitates the establishment of Li_2_O_2_ nano-architecture on the CNTs/Mo_2_C through ORR. The authors found that Mo_2_C/CNTs cathode exhibited 88% efficiency, whereas 74% efficiency for CNTs as well as for Mo_2_C powder. Furthermore, the polarization voltage of the Mo_2_C/CNTs was about 0.47 V, whereas it was 1.10 V for CNTs and 2.11 V for Mo_2_C. Overall, the Mo_2_C/CNTs cathode exhibited greater electrical efficiency along with low charge potential.

Hu et al. incorporated CNTs in ultrathin MoS_2_ nanosheets via a simple hydrothermal method and then used them as a cathodic catalyst in a Li-O_2_ battery [41]. Compared to the carbon and MoS_2,_ the MoS_2_/CNTs electrode showed a lower overpotential (nearly 0.29 and 1.05 V), satisfactory discharge capacity (6904 mA h g^−1^ at 200 mA g^−1^), greater capacity retention, the energy efficacy of about 79.17% and 64.5%, and increased cycle life (almost 132 cycles). The brilliant construction of MoS_2_/CNTs with superior electrical conductivity of CNTs and the high catalytic assets of MoS_2_ has enhanced the electrochemical efficiency of Li-O_2_ batteries. Wang et al. developed a green method for in situ encapsulating Co_2_P and Ru NPs on CNT (Co_2_P/Ru/CNT) and utilized it as a cathode electrode material in Li-O_2_ batteries [42]. Compared with Ru/CNT electrodes, Co/CNT and the Co_2_P/Ru/CNT electrode showed significantly enhanced ORR/OER as a result of the combined results of Ru and Co_2_P. Furthermore, the Li-O_2_ battery depends on the Co_2_P/Ru/CNT electrode amending ORR/OER typical overpotential of 0.75 V with an outstanding discharge/charge capacity about 12,800 mA h g^−1^ at 1 A g^−1^. Similarly, Pham et al. constructed a high-efficiency robust cathode for LOBs that relies on building a double-phase carbon nanoarchitecture integration of electroconductive CNTs and a porous structure metal organic framework (MOF) fabricated carbon (MOF-C) [165]. It is verified that the combined carbon architecture of the double-phasic MOF-C/CNT composite remarkably improved the electrochemical efficiency of LOBs (it had an excellent discharge capacity of nearly 10,050 mA h g^−1^ as well as a steady cycling efficiency above 75 cycles).

On the other hand, carbon nanofiber (CNF), carbon nano-cubes and N-doped mesoporous-activated carbon also showed remarkable electrochemical efficiency in Li-Air batteries. Nie et al. prepared hierarchically porous activated carbon nanofiber (ACNF) material and used them as a cathode electrode in a Li-O_2_ battery [166]. The results indicated that a Li-O_2_ battery along with ACNF cathodes exhibited improved discharge capacities (6099 mA h g^−1^) and decreased overpotential (2.75 V) in comparison to conventional BP 2000 composite (3538 mA h g^−1^) and non-activated CNF cathodes (4166 mA h g^−1^). These results demonstrated that the ACNF cathode in a Li-Air battery shows greater electrochemical performance. Sun et al. fabricated a mesoporous carbon nano-cubes nanoarchitecture with macropores and mesopores as cathodes for Li-O_2_ batteries are reported [167]. These materials showed a greater discharge capacity of about 26 100 mA h g^−1^ at a density potential of 200 mA g^−1^, better rate capability, and current density in comparison to a carbon black material. Zhu et al. produced N-incorporated activated carbon with a mesoporous structure (N-HMACs) using apples via a pyrolysis and carbonization strategy and the surface was modified with RuO_2_ NPs (N-HMACs-RuO_2_) [168]. The development of a Li-Air battery by using N-HMACs-RuO_2_ cathode electrode showed outstanding efficiency and a greater discharge capacity of about 13,400 mA h g^−1^.

Lately, carbon dots (CDs) or graphene quantum dots (GQDs) based nanocomposites as electrocatalysts and energy storage candidature have emerged as new types of quasi-zero-dimensional fluorescent carbon nanomaterials [169,170,171]. This material has unique properties, including numerous surface functional groups, tunable fluorescence emission, excellent water solubility, a high specific surface area, plentiful electron-hole pairs, variable heteroatom doping, electrical conductivity, rich electrochemical active sites, compatibility with numerous materials, and high stability [172,173,174,175,176,177,178]. The versatile CDs/GQDs can be used in combination with other active materials such as metal oxides for electrode materials including Li-Air batteries. They show enhanced specific capacity, cycle stability, and rate performance. Gao et al., proposed a new method to enhance the catalytic efficiency of CoO by the incorporation of CDs as well as oxygen vacancies [179]. Compared with commercial CoO with and without oxygen vacancies, the cyclic stability, starting stage capacity, in addition to rate performance of the prepared CoO/C electrode were highly improved, which is attributed to the synergistic outcome of CDs and oxygen vacancies on OER and ORR. The discharge capacity of the CoO/C-created electrode reached about 7000 mA h g^−1^ at the discharge potential of 100 mA g^−1^, demonstrating brilliant electrocatalytic activity. CDs can not only facilitate a great activity for ORR and steady oxygen vacancies during the process of either OER or ORR, but they also enhance the conductivity property of CoO. Wu et al. derived GQD from glucose via a hydrothermal process and demonstrated it as an effective cathodic electrode material in Li-Air batteries [180]. The GQD modified cathode showed a remarkable discharge capacity of about 68,900 mA h g^−1^ at 1400 mA g^−1^ current density. Furthermore, the GQDs-modified cathode presented excellent stability with the current density of 2000 mA g^−1^, and the capacity can still sustain 1000 mA h g^−1^ after 300 cycles. Lin et al. reported a laser-assisted fabrication of iron phthalocyanine/N-doped CDs incorporated on Co_3_O_4_ flakes (FePc/N-CDs@Co_3_O_4_) as a bifunctional electrocatalyst in OER and ORR [181]. FePc/N-CDs@Co_3_O_4_ as the cathode in the Li-O_2_ batteries demonstrates excellent stability at 1000 mA h g^−1^ for 350 cycles and a discharge capacity of 28,619 mA h g^−1^. The above-mentioned reports revealed that CDs have significant features to offer electron-hole pairs for improving conductivity, doping of the surface functional group/element, and the alteration of a surface functional group for increasing stability and electrochemical activity. The naturally renewable sources derived CDs or GQDs via green synthesis methods can be alternatives for other forms of carbon nanomaterials in Li-Air batteries. The aforementioned results indicated that the GQDs- or CDs-based materials were shown to be efficient electrode catalysts for Li-Air batteries. Moreover, CDs or GQDs and their nanocomposites are reported rarely in comparison with other carbon materials in energy storage applications. Therefore, it is interesting to prepare highly efficient CDs or GQDs and their nanocomposites-based electrodes for energy storage applications, including Li-ion batteries. Table 4 represents the comparative efficiencies of other carbon nanomaterials such as CNTs, CDs, GQDs and its composites.

The consolidated advantages of graphene as an electrode are as follows:Graphene has the highest surface sensitivity, optical transparency, chemical stability, and flexibility of any substance in the universe. It is also the strongest and thinnest material known to man. These qualities are good enough to make a material for electrodes [29].Several inorganic and organic materials can be doped or functionalized to create composite materials that give the catalyst a necessary property because of the high surface area and surface chemistry of graphene [30].Lithium ions can embed and de-embed more quickly in multilayer graphene materials because the space between layers is much wider than it is in graphite.Graphite is typically the most adaptable anode material. However, because of its high surface area, thermal and electrical conductivity, Young’s modulus, flexibility, and strength for use in flexible electrodes, etc., graphene might provide higher performance. Although it can hold more Li^+^, it is inactive while performing its function.Graphene has the flexibility and chemical inertness to operate as both a cathode and anode, providing outstanding performance in a variety of folded, curved, and reflattened geometries to inhibit the Li dendrites. Anode (Li) and cathode (NiCo_2_O_4_ and O_2_) are shielded with GO and rGO to prevent their deterioration or volume expansion to provide reliable performance with improved ORR and OER [38].The doped graphene can lower the overpotential and act as a cathode protective membrane and improve battery stability and longevity [72].Economical carbon materials such as graphene and CDs can be used in place of costlier noble metals and their oxides [87].The cost of the graphene electrodes can be minimized by using leftover graphite from old batteries and renewable sources in its preparation.Versatile morphology and dimensions with porosity and tunable ethylene and oxygenated carbons. These functional groups play crucial roles in the graphene chemical properties for many reactions by functionalizing a variety of molecules on top of it, both covalently and non-covalently [87].Graphene to rGO and GO band gaps are adjustable.Graphene can replace common catalysts such as Pt/C, since it possesses changeable energy levels and catalytic sites with dopants [102].Due to its exceptional hydrophobicity, chemical resistance, and high conductivity, graphene can prevent the growth of lithium dendrites at the anode and passivate the lithium anode to prevent interactions with moisture and oxygen [91].The graphene’s morphology, which ranges from a sheet-like structure to a porous one, is crucial in boosting the effectiveness of power densities [98].The lavish physical and chemical inertness of graphene was able to withstand the severe reaction circumstances and various electrode media, and can help to lower the overpotential.Owing to the aforementioned advantages, graphene electrode Li-O_2_ batteries are expected to have high charge speeds, stability, and a long life.

The disadvantages of graphene electrodes are as follows:The single layer graphene is expensive and requires a skilled technician to synthesize it in the lab.There is a chance of the reassembling the graphene monolayers into multilayers, which causes a decrease in the specific surface area for a catalytic reaction.Low-quality graphene may be prone to oxidation in a rich O_2_ environment. It may not provide the expected catalytic results and provide reversible capacity.The oxygen functional groups on GO may interact with the electrolytes and may cause overpotential.Physically adsorbed metals and metal oxides with graphene may cause the sintering/aggregation of the nanoparticles and may lower cell performance.

## 4. Challenges and Strategies

Several challenges exist in the battery industries and with regard to Li-Air batteries. The most common current challenges are safety issues [182,183] and the recycling of used batteries [184]. Their efficacy, fast charging ability, durability, heat resistance, weight, and economic availability have to be improved further. Apart from general challenges, specific problems of Li-Air batteries have been systematically projected in Figure 9 [184]. Figure 9a illustrates the Li-Breathing cells’ targeted power densities and voltages that have to be reached by overcoming the problems associated with every component very carefully. The creation of various kinds of catalysts, such as porous electrode materials (C, B, N, S, metal, and metal oxides), and stable electrolyte solutions have helped to reduce the inherent difficulties associated with O_2_ chemistry while breathing. At the following stage, we must reconstruct batteries by using air from the Earth’s atmosphere in place of pure O_2_ gas. Therefore, it is essential to properly address the major issues that Li-Air batteries are currently facing, which include the selective filtration of O_2_ from the air and the suppression of unfavorable reactions with other airborne components such as CO_2_, N_2_, and H_2_O vapor. Figure 9B displays all crucial elements for creating Li-Air batteries that are optimized for real time use at ambient conditions. Hence, we have discussed all possible reported battery components in detail along with cathode materials based on graphene and other carbon materials. To improve the carbon electrode performance in the Li-Air battery the following points have to be optimized: (1) fabricating with a high surface area; (2) increasing active sites of the carbon skeleton (increase edges and pores); (3) doping, co-doping and fictionalization with metals and metal oxides; (4) creating macro pores to enhance the diffusion of O_2_ from the air; (5) identifying the discharge products and analysis; and (6) finding the economic sources of the catalyst fabrication to lower the price. To some extent, the above parameters have been optimized. However, there remains a lot to work in this regard by a variety of combinations of electrolytes, separators, RMs, and membranes to protect the electrodes to achieve more reliable specific capacity and stability [185,186,187,188,189].

## 5. Conclusions and Future Perspectives

In summary, the Li-Air cell was composed of a Li metal anode and an O_2_ cathode made up of porous carbon, GO, rGO, CNTs, CNFs, GQDs and other carbons-supported metals and metal oxides. As the focus of this review is on the cathode materials based on graphene, we have comparatively discussed graphene, other carbon materials, doped graphene with B, N, S, Ru and other metal oxides. According to our survey, we found that porous graphene and GQDs have shown remarkably high specific capacities than other graphene materials such as CNTs, non-metal, metal, and metal oxides doped/supported graphene catalysts. The discharge/charging process due to ORR and OER at the cathode is another key focus of this review. Both the reaction kinetics are facile at better reaction conditions at real-time demonstration, when the cell is fabricated with superior cell components.

An ideal construction of a battery could be possible by a thorough understanding of the materials. The physical, electrochemical, structural properties, and theoretical studies can provide the rational design of the battery composites. As the true Li-Air has not been greatly commercialized and is still under practical and experimental investigation and demonstration, ample work has to be done in the improvising of all battery components such as anodes, cathodes, electrodes, electrolytes, separators, membranes, and packing materials. This will be realized when all research experts integrate their efforts and knowledge in a convergent direction for the safe, efficient, economical Li-Air batteries in the market with good recycling technologies. Furthermore, the facile synthesis of graphene-related materials, their exceptional physical and chemical stability, outstanding conductivity, surface area, flexibility, transparency, eco-friendliness, and availability from cheap resources made it to study extensively for the ORR and OER catalytic reactions in Li-Air batteries. Further improvements in the fabrication of highly stable, conductive and optimal doping porous graphene are anticipatively demanded to improve the efficacy of the battery along with the cost. As experts in graphene, we suggest preparing the graphene, GO, and rGO from renewable sources with high purity, surface area, and desired morphology to impart high conductivity. This can reduce the electrode cost and weight of the battery. Good experimental skills are needed to create selective defects, 3D architectures, and doping of non-metals and metals to improve the catalytic ability and stability. The amount of active sites, defects, and doping levels has to be critically calculated and determined. Binder-free methods are highly encouraged to deposit the catalyst so as to minimize the chemicals, and extra expenses are suggested. As most of the experiments are not done in the presence of air and are done in real-time, an accurate assessment of the existing literature might exaggerate the findings. A greater number of real-time demonstrations based on graphene, doped graphene, and GQDs-based cathodes in Li-O_2_ batteries are important to access its ability of stable energy production.

## Data Availability

Not applicable.

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
