# Peer review of "Status Quo on Graphene Electrode Catalysts for Improved Oxygen Reduction and Evolution Reactions in Li-Air Batteries"

_molecules, 2022, doi:10.3390/molecules27227851_

Round 1

Reviewer 1 Report

Review article submitted by Yong-Chien Ling regards the application of graphene in Li-O2 batteries. Article could be of interest for the readers of the journal but prior to its publication I suggest some modifications in order to improve the impact and readability o the paper.

In order to improve the impact of the review I suggest to the authors to reduce the first two sections, as well as the first part of conclusion, of the paper not strictly related to the application of graphene in Li-O2 batteries.

Section 3.1: Production of graphene starting from end-of-life lithium-ion batteries should be included in this section 10.1021/acssuschemeng.1c04690). This kind of graphene production should be discussed in order to emphasize that this new promising class of Li battery (Li-O2) could be produced starting from waste Li-ion batteries, thus limiting the consumption of primary source.

Author Response

Please see the detail in the attachment.

Reviewer 2 Report

The Li-air battery is a hot research topic. This manuscript reviewed the challenge and advantages of graphene-based Li-O2 batteries. This paper is well-structured, and timely. I think it is worth being published. However, some issues should be addressed.

1. Some figures are not clear. The authors are suggested to upload high-resolution figures.

2. The mechanism of improving the electrochemical performance by B, N, and S doping has not been explained clearly in this manuscript. The authors should add it.

3. Oxygen functional groups on graphene should be discussed in this manuscript (such as C−O (ether, hydroxyl, and epoxy), C=O (carbonyl and quinone), –COO– (carboxyl and ester), and so on).

4. Do the rGO/Ru/aMnO2 electrode prepared by vacuum filtration method mentioned in the “Graphene-Metals” section also belong to the type of “Graphene-Metal Oxides”?

5. The title of the “Challenges” is suggested to be changed to “Challenges and Strategies”. (page 26)

6. The perspective of future research is not clear enough to guide the development of this field. The advantages and disadvantages of various graphene electrodes were not discussed in this review.
7. some associated references (Rare Metals. 2021;40(11):3156–3165;
Electrochimica, Acta, 2020, 330: 135119.;Rare Metals. 2021,40(10):2657–2689;Journal of Energy Chemistry, 2018, 27(2): 419-425; Electrochimica, Acta, 2018, 265: 577-585; Rare Metals. 2022;41(1):125–131.) should be cited to further emphasize the novelty of this work.

Author Response

(The authors gave the same response as above.)

Reviewer 3 Report

In this manuscript, Gollavelli et al. provided a comprehensive literature review on the applications of Graphene electrode catalysts for improved oxygen reduction and evolution reactions (ORR/OER) in Li-air batteries. The review paper covers various aspects, including fundamentals of Li-air battery and its key components, graphene materials and their composites as ORR/OER catalysts. Overall, the paper provided a balanced overview and summary of literature with well-organized display items to serve this purpose. This reviewer would like to support its publication in the Molecules journal pending minor revision taking into consideration the below comments.

1. The Abstract did not mention much about Graphene electrode catalysts. It is suggested that some of the advantages/characteristics/features of graphene electrode catalysts should be introduced in Abstract.

2. In Abstract, the “Due to the oxygen reduction reaction and oxygen evaluation reaction, the anode and cathode usually face many problems” sentence needs to be revised. (1) it should be oxygen evolution and not oxygen evaluation. (2) the oxygen reduction and evolution is for the cathode and not anode.

3. Many writing problems can be found in the manuscript, including typos and grammar errors, which must be corrected before acceptance. For instance, (1) line 35, “full fill”. (2) line 29, the first sentence of Introduction was grammatically incorrect and difficult to understand. (3) Line 892, “OES”. Also please double check the writing throughout the manuscript.

4. To appeal to a broader readership, recent works on Li-O2 batteries (Journal of Materials Chemistry A 2015, 3, 16132-16141) and Zn-air batteries (Small, 2022, 18, 2105329; https://doi.org/10.1002/9783527837939.ch3) are suggested to be referenced in Introduction.

5. Figures need to be improved. For example, Figure 1a, bottom line, “LiCO2” should be revised into “Li-CO2”.

6. In Scheme 1, the wording of the bullet points for Advantages and Disadvantages need to be revised. (1) “Over potentials” should be revised into “Overpotentials”. (2) The first letter of “Power” and “Stability” should not be capitals for consistency with other words. (3) Both nouns and adjectives were used, which are not consistent.

7. In section “6. Conclusions and Future Perspectives”, more discussion should be made regarding the future perspectives. The current manuscript lacks the authors’ own perspectives in graphene electrode catalysts.

Author Response

(The authors gave the same response as above.)

Round 2

Reviewer 1 Report

Suggestions were addressed, the paper can be now published.

Reviewer 2 Report

Accept in present form.